# ACTIVE PREFERENCE OPTIMIZATION VIA MAXIMIZING LEARNING CAPACITY

## ABSTRACT

The success of deep learning in various complex tasks relies heavily on large amounts of annotated data, which can be prohibitively expensive to acquire. Techniques such as reinforcement learning with human feedback (RLHF) and direct preference optimization (DPO) have emerged as methods for fine-tuning models by leveraging human preferences, but they come with significant costs, especially when applied to large-scale language models (LLMs). Recent efforts to reduce these costs have focused on active preference optimization, which uses certainty-based selection to minimize the annotation burden. However, the two-step process of selecting uncertain input prompts and then acquiring completions can lead to suboptimal pairings, potentially limiting model learning capacity. This paper suggests that divAPO eliminates suboptimal pairings that are typical of two-step methods and enhances learning capacity by selecting the most informative preference pairs in a single phase, taking into account both data distribution probabilities and preference model certainty. Through experiments on complicated Language tasks, we demonstrate that our method achieves significant performance improvements over existing approaches.

## 1 INTRODUCTION

The success of deep learning in a variety of intricate tasks is significantly influenced by the availability of extensive, well-annotated data, which can be prohibitively expensive to acquire in practice. Reinforcement learning with human feedback (RLHF) (Ouyang et al., 2022) and direct preference optimization (DPO) (Rafailov et al., 2024) are two techniques that have emerged in recent years as approaches to fine-tuning models by leveraging human preferences. Nevertheless, these methods result in substantial expenses, particularly when fine-tuning large models such as GPT (Brown, 2020) or diffusion models (Ho et al., 2020), which necessitate a significant number of responses from human participants with domain expertise, as well as large-scale language models (LLMs). In addition, the generation of numerous responses for a single input prompt and their subsequent ranking are typical characteristics of preference labels, necessitating a substantial quantity of responses (Ouyang et al., 2022; Jiang et al., 2024; Ziegler et al., 2019; Liu et al., 2020). Nevertheless, it is imperative to identify methods to optimize the data acquisition process, as the cost of preference labeling from all of these responses can be substantial.

In this regard, to reduce these costs, recent research has explored the integration of active learning (AL) strategies, such as active preference learning (APL) (Muldrew et al., 2024). APL suggests a certainty-based selection strategy for correcting the overconfident prediction of DPO. According to Figure 1 (a), this method involves a two-step selection process. First, predictive uncertainty calculates the average entropy score of the output text token probability to identify informative input prompts. Then, preference model certainty—which is defined using the implicit rewards gap of the preference model, like DPO—acquires corresponding completions. These strategies aim to minimize the annotation burden by intelligently selecting the most informative input prompt and completion pairs for labeling.

Although this strategy effectively selects uncertain input prompts, it fails to consider the potential informativeness of the input prompt and completion pairings from a fine-tuning perspective, which could restrict the model's overall learning capacity. Furthermore, no assurance selecting an input prompt based on its uncertainty will produce a high-quality or well-informative completion pair,

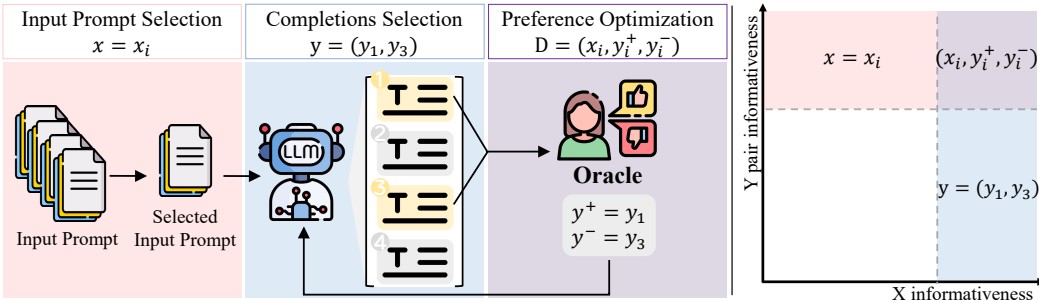

(a) Active Preference Optimization with 2-step selection          (b) Learning capacity of $x$-$y$ pair

Figure 1: Overview of active preference optimization; (a) For selected input prompt $x = x_i$ and completions $y = (y_1, y_2, ...)$, the oracle then ranks preference as $y^+$ (preferred) and $y^-$ (not preferred). (b) Learning capacity of the input prompt - completions pair, where the x-axis and y-axis represent the informativeness of the input prompt and the completions pair.

which could result in suboptimal pairs for model learning. For instance, input prompts that contain philosophical content are more likely to be chosen due to their ambiguity, even though they are less frequently encountered in everyday use. Nevertheless, the probability of the model encountering difficulty in comprehending these prompts is elevated, resulting in summaries of inferior quality. The x-axis in Figure 1 (b) represents the informativeness of the input prompt, such as uncertainty. The blue region denotes the critical portion that the input prompt has selected. The informativeness of completion pairs is represented by the y-axis, while the magenta region denotes the critical portion that the completion pairs have selected. In active learning with preference optimization, the purple portion is the most critical subset that we are striving to identify. In the 2-step selection, certain informative pairs (shown in the pink region) are not selected under this strategy, even though they hold a high value for fine-tuning from a broader perspective. This observation emphasizes the necessity of an active learning strategy that simultaneously prioritizes both input prompts and completion pairs.

In this paper, we present divAPO, a novel active learning strategy that is intended to enhance preference learning by simultaneously evaluating the informativeness of the pairs and the diversity of input prompt stimuli. Our approach considers both the data distribution probabilities and preference model certainty regarding the completions pairs, allowing for a more comprehensive selection of informative samples. Our method employs an input prompt-completions one-step selection strategy, eliminating the sub-optimal candidate selection issue that arises when either input prompt or completions is chosen first. We were able to select data more diversely by approximating the distribution of input prompts and completions and incorporating the model's assurance to modify this distribution. By employing this approach, we were able to identify an optimal subset that effectively balances diversity and informativeness, thereby improving the overall learning process. Furthermore, we underscore the significance of input prompt diversity, as the repetitive sampling of similar data points resulting from multiple overlapping input prompts and completion configurations within the dataset can result from the selection of uncertain data alone.

We validate our approach through extensive experiments on various tasks, including IMDB movie review generation, TL;DR text summarization, and Anthropic HH (helpfulness-harmlessness) single-turn dialogue. Using large-scale, open-source LLMs with approximately 1–3 billion parameters, our results demonstrate a significant performance improvement, with an increase up to 9.21% over baseline methods. These findings underscore the effectiveness of our proposed method in leveraging preference feedback for AL, contributing to both the theoretical understanding and practical application of AL in the context of modern machine learning tasks. The source code is available at https://anonymous.4open.science/r/divAPO-8A96.

## 2 RELATED WORK

### 2.1 PREFERENCE OPTIMIZATION

The rapid advancements in Large Language Models (LLMs) have highlighted the importance of aligning models with intricate human preferences (Jiang et al., 2024). Preference optimization (PO)

has emerged as a promising technique in fine-tuning these models, leading to a compelling model such as ChatGPT (Ouyang et al., 2022). One of the pioneer approaches in this domain is Reinforcement Learning from Human Feedback (RLHF) (Ouyang et al., 2022) that involves a two-stage process: 1) learning an auxiliary reward model on human preference datasets, and 2) fine-tuning LLMs to align the reward with RL objectives, such as Proximal Policy Optimization (PPO) (Schulman, 2017), P3O (Wu et al., 2023), and REINFORCE (Ahmadian et al., 2024).

As many studies have reported that RLHF often suffers from high variance and sensitivity to hyperparameters, Direct Preference Optimization (DPO) (Rafailov et al., 2024) has emerged as an alternative by not requiring explicit reward modeling or RL training. DPO simplifies the training process by directly maximizing the output likelihood of preferred data examples while reducing that of non-preferred examples. Several variants of supervised loss of preference optimization have been studied (Azar et al., 2024). SLiC-HF (Zhao et al., 2023) introduces a hinge-loss objective to further separate preferred and non-preferred samples. IPO (Azar et al., 2024) replaces the traditional sigmoid-based objective with a mean squared error (MSE) loss to enhance training stability. Despite their promising results, the sample selection problem for constructing better preference datasets remains underexplored, yet is crucial for further refining alignment processes.

## 2.2 ACTIVE LEANING FOR PREFERENCE OPTIMIZATION

**Active Learning in NLP.** AL is a well-established problem with a primary goal to maximize model performance with minimal labeling costs by identifying the most informative samples and labeling them with human oracle (Zhang et al., 2022; Cohn et al., 1994; Houlsby et al., 2011; Settles, 2009). In NLP, there are two main traditional AL strategies commonly used (Zhang et al., 2022). One strategy is based on the informativeness of data instances, utilizing factors such as the uncertainty and disagreement of the models (Engelson & Dagan, 1996; Huang et al., 2024; Siddhant & Lipton, 2018). The other strategy is based on the representativeness of data instances (McCallum et al., 1998; Settles & Craven, 2008; Zhao et al., 2020), which focuses on selecting representative samples that can effectively capture the diversity of the underlying data distribution (Bloodgood & Callison-Burch, 2014; Eck et al., 2005; Zeng et al., 2019). These two directions are particularly necessary for complex tasks including text generation or summarization, where the selected samples need to be both informative (Zhao et al., 2020; Gal & Ghahramani, 2016; Gidiotis & Tsoumakas, 2023) and representative (Sener & Savarese, 2017; Tsvigun et al., 2023; Perlitz et al., 2023). However, most AL methods have traditionally focused on supervised learning scenarios, where only a single label is required for annotating each data example, making them hard to directly apply to the PO framework, which usually requires a pair of answers (or labels) to incorporate the human preference.

**Active Preference Optimization.** Recently, a few approaches have attempted to study active learning for preference optimization. APL (Muldrew et al., 2024) particularly involves a *two-step* sample selection process based on the sample uncertainty and DPO reward; it first roughly selects samples with a high LLM output entropy, and then refines the choices by further selecting samples with a high DPO reward gap between a pair of answers. While APL shows enhanced sample efficiency with the two-step uncertainty-perspective selection, it lacks studies of the effect of diversity and does not validate generalizability over various PO approaches, such as SLiC-HF.

## 3 PROBLEM STATEMENT

### 3.1 PRELIMINARY

Let $\mathcal{D}_p = \{(x, y^+, y^-)\}$ be a preference dataset of LLMs, where $x$ denotes an input instruction and $(y^+, y^-)$ denotes its pairwise answer annotated by human as $y^+$ is more preferable than $y^-$ to answer the instruction $x$.

### 3.2 ACTIVE LEARNING FOR PREFERENCE OPTIMIZATION

Let $\theta$ be an initial LLM trained on general text corpus with SFT, and $U = \{x_i\}_{i=1}^m$ be an unlabeled instruction dataset for the target domain. To annotate this dataset with human preference, a human

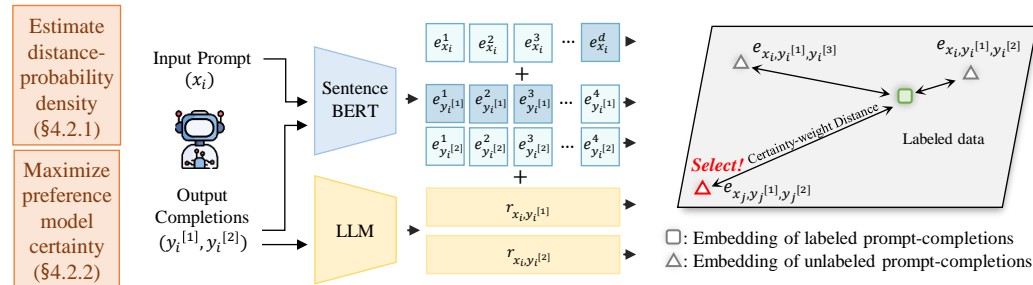

Figure 2: Overview of the divAPO. First, the distance-probability density is estimated (Section 4.2.1), followed by maximizing the preference model certainty (Section 4.2.2). The process begins with the selection of an input prompt and output completions generated by a large language model (LLM) trained with preference feedback. The embeddings of labeled and unlabeled input prompt and completions are compared to find the optimal completion pair.

oracle is requested to assign a preference label between a pairwise answer $(y^{[1]}, y^{[2]})^1$. If the human prefers the answer $y^{[1]}$ over $y^{[2]}$ then the annotated preference answer $(y^+, y^-) = (y^{[1]}, y^{[2]})$, otherwise $(y^+, y^-) = (y^{[2]}, y^{[1]})$. Then, a problem of *active learning for preference optimization* aims to find the most informative query set $S \subset U$, that is augmented to the preference labeled set $L = \mathcal{O}(S)$ by an oracle $\mathcal{O}$, within an annotation budget $b$ by the following objective:

$$S^* = \underset{S \subseteq U : |S| \leq b}{\text{argmax}} \; \text{Alignment}(\hat{\theta}) : \hat{\theta} = \underset{\theta}{\text{argmin}} \sum_{(x, y^+, y^-) \in \mathcal{O}(S)} \mathcal{L}_{pref}(x, y^+, y^-; \theta), \quad (1)$$

where Alignment$(\cdot)$ is the model's alignment performance to human preference. This AL for PO problem can be extended into the multi-round AL scenario by simply accumulating the labeled set $L$ throughout the AL rounds.

### 3.3 PREFERENCE MODEL CERTAINTY

With characteristics of the Bradley-Terry model, DPO has the implicit reward function $\hat{r}(x, y) = \beta \log \frac{p_\theta(y|x)}{p_{\theta_{ref}}(y|x)}$ where $\beta$ is a hyperparameter that controls the proximity to the SFT model $\theta_{\text{ref}}$. The preference model certainty $R$ (Muldrew et al., 2024) for a data point $x_i$ is defined based on the absolute difference between the model's predictions for two responses $y_i^{[1]}$ and $y_i^{[1]}$:

$$R(x_i, y_i^{[1]}, y_i^{[2]}) = \left| \hat{r}(x_i, y_i^{[1]}) - \hat{r}(x_i, y_i^{[2]}) \right|. \quad (2)$$

Data points with larger discrepancies can drive significant learning, especially when the model's confident predictions differ from the oracle's evaluation. These discrepancies are crucial for improving the model's performance. This aligns with the DPO training objective, which emphasizes gradient updates based on prediction deviations, allowing the model to focus on correcting its most notable errors while the KL constraint ensures it stays close to its previous behavior.

## 4 DIVAPO

### 4.1 OBJECTIVE

We propose divAPO, a method for efficiently learning preferences with a limited budget on preference annotations. By using a 1-step selection process, divAPO reduces suboptimality and directly maximizes the objective for improved performance. The goal is to select a subset $\mathcal{S}$ to maximize the preference model certainty, denoted as $R(\cdot)$, over the entire data distribution $P(U)$. The inclusion of $P(U)$ directly into the objective function enables our method better to reflect the underlying

---

[1]This pairwise answer can be produced by the initial LLM $\theta$ or can be obtained by humans.

distributional characteristics of the data. The objective function can be mathematically expressed as follows:

$$\mathcal{S}^* = \underset{\mathcal{S} \subseteq U : |\mathcal{S}| = b}{\arg\max} \sum_{x_i \in \mathcal{S}} \left( P(x_i) \cdot R(x_i, y_i^{[1]}, y_i^{[2]}) \right) \tag{3}$$

## 4.2 DIVAPO

In practice, the exact data distribution $P(U)$ is unknown and cannot be directly computed. Therefore, we introduce an approximated objective based on a distance-estimated probability (DEP) distribution that effectively captures the data's spatial properties. Using Definition 4.1 and Equation 6, the objective function can be expressed as follows:

$$\mathcal{S}^* = \underset{\mathcal{S} \subseteq U : |\mathcal{S}| = b}{\arg\max} \sum_{x_i \in \mathcal{S}} \left( P_d(x^i, L) \cdot R(x_i, y_i^{[1]}, y_i^{[2]})) \right) \tag{4}$$

This formulation ensures that the selected subset maximizes the overall expected preference model certainty while considering the underlying data distribution via the distance estimated probability. In Section 4.2.1, we first introduce a new metric, the distance estimated probability, that enables estimating the $P(U)$. Then, in Section 4.2.2, we introduce preference model certainty, which represents learning capacity of preference optimization.

### 4.2.1 DISTANCE ESTIMATED PROBABILITY

We approximate $P(U)$ using a distance-based probability metric, which we refer to as the distance-estimated probability (DEP). This approximation is efficient for modeling distributions and facilitates its use in our objective function. Formally, DEP is defined as follows:

**Definition 4.1.** (DISTANCE ESTIMATED PROBABILITY). The distance estimated probability $P_d(x_i, L)$ of instance $x^i$ and labeled set $L$ is formalized as

$$P_d(x_i, L) = \frac{d_p(e(x_i, y_i^{[1]}, y_i^{[2]}), L)}{\sum_{x_i \in U} d_p(e(x_i, y_i^{[1]}, y_i^{[2]}), L)} \tag{5}$$

Here, $e(x^i, y_1^i, y_2^i)$ is the embedding of point $x^i$ with corresponding features $y_1^i$ and $y_2^i$, $d_p(\cdot)$ represents the distance (Arthur & Vassilvitskii, 2006), can be based on various distance metrics (e.g., Euclidean, cosine).

### 4.2.2 MAXIMIXE EXPEXTED PREFERENCE MODEL CERTAINTY

As mentioned in the Section 3.3, the preference model certainty can be defined with the reward gap of the implicit preference model.

$$R(x_i, y_i^{[1]}, y_i^{[2]}) = \left| \hat{r}(x_i, y_i^{[1]}) - \hat{r}(x_i, y_i^{[2]}) \right| \tag{6}$$

Data points with larger discrepancies can drive significant learning, especially when the model's confident predictions differ from the oracle's evaluation. These discrepancies are crucial for improving the model's performance. This aligns with the DPO training objective, which emphasizes gradient updates based on prediction deviations, allowing the model to focus on correcting its most notable errors while the KL constraint ensures it stays close to its previous behavior.

Once the data distribution has been estimated, we incorporate the preference model certainty for each data point to adjust the sampling probabilities. The expected preference model certainty for a subset of the data is computed by multiplying the distance estimated probability-derived probability by the preference model certainty of each instance in the subset. We normalize the preference model certainty across the dataset and apply a weighting factor to emphasize the importance of certain instances based on their relative distance and contribution to the learning process.

## 4.3 Greedy Selection Algorithm

Given that our objective function, which incorporates DEP and the preference model certainty, is NP-hard, solving it directly is computationally intractable. As a result, we employ a greedy selection algorithm to approximate the optimal solution. The objective function satisfies the properties of monotonicity and submodularity, which ensures that the return of the function increases monotonically as new examples are added to the subset. Moreover, the marginal benefit of adding a new example diminishes as the subset grows. These properties justify the use of a greedy algorithm, as the solution is guaranteed to be within a constant factor of the optimal solution, as demonstrated in A.

In Theorem 4.2, we guarantee the selected subset $\mathcal{S}$ obtained by our greedy solution achieves a $(1 - 1/e)$-approximation of the optimum.

**Theorem 4.2.** *Since Eq. (4), denoted as $OBJ$, is a monotone, submodular, and non-negative function on x, the greedy solution provides a set with a $(1 - 1/e)$-approximation of the optimum. Formally,*

$$OBJ(\mathcal{S}) \geq (1 - 1/e) \cdot OBJ(\mathcal{S}^*). \tag{7}$$

*Proof.* We prove the monotonicity and submodularity of Eq. (4). If the two conditions are satisfied, Eq. (7) naturally holds. See Appendix A for the complete proof. □

The k-means++ seeding algorithm inspires our greedy selection process. We iteratively select data points by measuring the p-norm distance between the unlabeled preference data points and the currently selected subset $\mathcal{S}$. For each unlabeled data point, we compute its distance to the nearest data point in $\mathcal{S}$, and this distance is used to derive a sampling probability. At each step, we compute the DEP for the unlabeled data points and multiply this by their normalized preference model certainty. The data point with the highest combined score is then selected and added to the subset $\mathcal{S}$. The greedy sample selection can be employed as in Algorithm 1.

---

**Algorithm 1** Greedy Selection Algorithm of divAPO

INPUT: Unlabeld Data set $U = \{x_i\}_{i=1}^{m}$, Target subset size $s$, Embedding function $e$, Initial model $\theta_0$, Oracle $\mathcal{O}$, Active learning round $R$
OUTPUT: Optimal subset $\mathcal{S}^*$
1: $S \leftarrow \emptyset$
2: $\theta_t \leftarrow \theta_0$
3: $\mathcal{X}_p := \{x_i, y_i^{[1]}, ..., y_i^{[n]}\}_{i=1}^{m} \leftarrow$ Generate$(\theta_t, U)$
4: **for** $r \leftarrow 1$ to $R$ **do**
5:   **repeat**
6:     $d_p(e(x_i, y_i^{[1]}, y_i^{[2]}), L) \leftarrow \min_{x_j \in L} \left( \sum_k |e(x_i, y_i^{[1]}, y_i^{[2]}) - e(x_j, y_j^{[1]}, y_j^{[2]})|^p \right)^{\frac{1}{p}}$
7:     $P_d(x_i, L) \leftarrow d_p(e(x_i, y_i^{[1]}, y_i^{[2]}), L) / \sum_{x_i \in U} d_p(e(x_i, y_i^{[1]}, y_i^{[2]}), L)$
8:     $R(x^i, y_1^i, y_2^i) \leftarrow \left| \hat{r}(x_i, y_i^{[1]}) - \hat{r}(x_i, y_i^{[2]}) \right|$
9:     $\{x_*, y_*^+, y_*^-\} \leftarrow \arg \max_{x' \in U \backslash \mathcal{S}} \left( P_d(x^i, e) \cdot R(x^i, y_1^i, y_2^i) \right)$
10:   **until** $|\mathcal{S}| = s$
11:   $\mathcal{S} \leftarrow \mathcal{S} \cup \{x_*, y_*^+, y_*^-\}, \mathcal{X}_p \leftarrow \mathcal{X}_p - \mathcal{S}$
12:   $\theta_{t+1} \leftarrow$ Finetune$(\theta_0, \theta_t, \mathcal{S})$

---

**Time Complexity Analysis.** The time complexity of the divAPO strategy can be understood by breaking down its key components. The primary computational steps include embedding extraction, reward calculation, and diverse point selection using a clustering-based approach. First, embeddings are calculated for all samples, which requires $O(m \cdot d)$, where $m$ is the number of unlabeled data points and $d$ is the dimensionality of embeddings. Next, rewards for each sample are computed with complexity $O(m)$. To calculate distance estimated probability, we use the p-norm distance between embeddings, with complexity $O(m \cdot k \cdot d)$, where $k$ is the number of querys. Overall, the dominant factor is the clustering process, yielding an overall time complexity of $O(m \cdot k \cdot d)$ per query iteration.

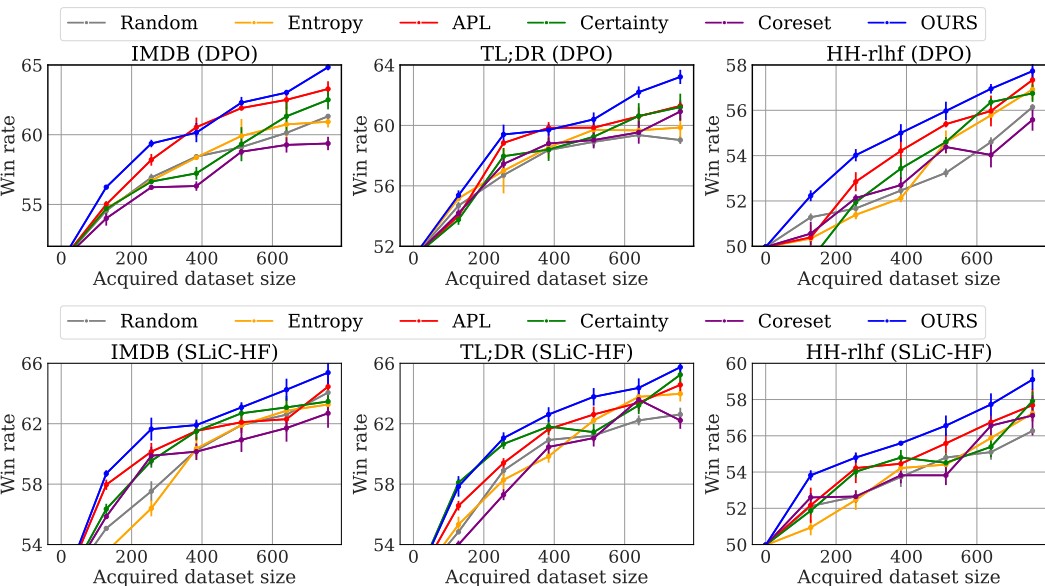

Figure 3: Win-rate comparison using the DPO and SLiC-HF loss function. (a) IMDB shows win-rate versus human-provided positive review data of test set; (b) TL;DR presents win-rate versus human-provided summaries on test prompts; (c) HH-rlhf displays win-rate versus the initial model. The x-axis represents the size of the acquired dataset used for fine-tuning at the evaluation point.

## 5 EXPERIMENTS

### 5.1 EXPERIMENT SETTING

**Tasks and Datasets.** We evaluated LLM alignment performance across several datasets, including IMDB (Maas et al., 2011), TL;DR (Völske et al., 2017), and HH-RLHF (Bai et al., 2022). IMDB consists of 25K movie reviews, and the task focuses on continuously generating positive reviews from a short initial review. TL;DR contains 117K Reddit posts, and the task is to generate summaries of those posts. HH-RLHF consists of 170K dialogues between humans and automated assistants, and the task aims to increase the assistant answer's helpfulness while decreasing its harmlessness.

**Algorithms.** We compare divAPO with a random sample selection, Uniform, and *three* conventional AL approaches for supervised learning, Entropy (Kadavath et al., 2022), Certainty (Muldrew et al., 2024), and Coreset (Sener & Savarese, 2017), and *one* AL approach for preference optimization, APL (Muldrew et al., 2024). Entropy selects samples with a high average entropy score of the output text token probability. Certainty selects the pair completions where the model's implicit reward value is high. Coreset selects a subset of examples that maximize the distance coverage to the entire training set in the embedding space of SentenceBert (Reimers, 2019). APL performs sample selection by a two-step uncertainty-based filtering, that first roughly selects samples with a high LLM output entropy, and refines the choices by further selecting samples with a high DPO reward gap.

**Implementation Details.** We validate our experiment using two promising preference optimization approaches, DPO and SLiC-HF. For the initial backbone LLM, we employ GPT-2 (Radford et al., 2019) for IMDB dataset, Pythia-1B (Biderman et al., 2023) for TL;DR dataset, and Pythia-2.8B for HH-RLHF dataset. All the backbone LLMs are trained by SFT first, and then used for preference optimization. For the experiments, we followed the hyperparameter setup used in (Muldrew et al., 2024), utilizing the ADAM optimizer with a learning rate 1e-06. We trained 30 epochs for IMDB data and 50 epochs for TL;DR and HH-rlhf data. A batch size of 32 was employed, and each query round included a query size of 128 samples with a total of 6 rounds. Additionally, we set the hyperparameter $\beta$ to 0.2. To calculate embeddings for divAPO, we use the same sentence embedding model, all-MiniLM-L6-v2 SentenceBERT, with CoreSet. We repeated every experiment 5 times and reported the average value. All experiments were conducted using a 48GB A6000 GPU. Further implementation details can be found in Appendix B.

| PO | DPO | | | SLiC-HF | | |
|---|---|---|---|---|---|---|
| **Data** | **IMDB** | **TL;DR** | **HH-rlhf** | **IMDB** | **TL;DR** | **HH-rlhf** |
| **Uniform** | $61.32\pm_{0.13}$ | $59.03\pm_{0.26}$ | $56.14\pm_{0.13}$ | $64.06\pm_{0.27}$ | $62.61\pm_{0.45}$ | $56.27\pm_{0.13}$ |
| **Entropy** | $60.93\pm_{0.41}$ | $59.86\pm_{0.67}$ | $56.92\pm_{0.48}$ | $63.27\pm_{0.71}$ | $63.98\pm_{0.51}$ | $57.24\pm_{0.14}$ |
| **Certainty** | $62.50\pm_{0.68}$ | $61.21\pm_{0.89}$ | $56.75\pm_{0.38}$ | $63.47\pm_{0.61}$ | $65.24\pm_{0.54}$ | $57.92\pm_{0.52}$ |
| **Coreset** | $59.37\pm_{0.48}$ | $60.92\pm_{0.60}$ | $55.58\pm_{0.48}$ | $62.69\pm_{0.96}$ | $62.22\pm_{0.56}$ | $57.12\pm_{1.1}$ |
| **APL** | $63.28\pm_{0.55}$ | $61.29\pm_{0.38}$ | $57.34\pm_{0.55}$ | $64.45\pm_{0.82}$ | $64.57\pm_{0.52}$ | $57.69\pm_{0.90}$ |
| **OURS** | $\mathbf{64.84}\pm_{\mathbf{0.22}}$ | $\mathbf{63.21}\pm_{\mathbf{0.47}}$ | $\mathbf{57.73}\pm_{\mathbf{0.32}}$ | $\mathbf{65.38}\pm_{\mathbf{0.91}}$ | $\mathbf{65.74}\pm_{\mathbf{0.58}}$ | $\mathbf{59.10}\pm_{\mathbf{0.56}}$ |

Table 1: Win-rate performance comparison across IMDB, TL;DR, and HH-rlhf datasets with DPO and SLiC-HF loss functions. We calculated the mean and standard deviation of three experimental runs from the most recent round's win rate to determine each result.

**Evaluation.** We use GPT-4o-mini as the oracle for evaluating preference, based on many studies indicating that OpenAI's GPT-4, when appropriately prompted, aligns closely with human judgments (Rafailov et al., 2024; Muldrew et al., 2024). We further validate that its recent cost-efficient variant, GPT-4o-mini, also provides consistent and high-quality responses. See Appendix C for more detailed explanations. Therefore, we prompt GPT-4o-mini to evaluate the LLM response on three criteria: (1) relevance to the task, (2) grammatical accuracy, and (3) sentence consistency. Given these detailed score rubrics, GPT-4o-Mini determines which model's output response is more preferred, with the final metric referred to as "win-rate". We present the prompt in Appendix D.

## 5.2 MAIN RESULTS

Figures 3, along with Table 1, present the performance comparison of various active learning strategies across the IMDB, TL;DR, and HH-rlhf datasets using the DPO and SLiC-HF loss functions. The results demonstrate that our proposed method, divAPO, consistently outperforms the baseline approaches—Random, Entropy, APL, Certainty, and Coreset—across all datasets and loss functions.

Across all datasets and configurations, Figure 3 shows our proposed method, divAPO, consistently outperforms the baseline approaches—Random, Entropy, APL, Certainty, and Coreset. In the IMDB dataset, divAPO achieves the highest win-rate in both the DPO and SLiC-HF settings, demonstrating its ability to effectively leverage informative samples as the dataset size increases. The trend is similar for the TL;DR and HH-rlhf datasets, where divAPO shows the most significant improvements in performance, particularly as the acquired dataset size grows. These results highlight the superiority of our method in maximizing model performance by balancing preference optimization and diversity within the selected samples, outperforming the baselines in all cases.

Table 1 summarizes the win-rate results from the last round of the experiments, calculated as the mean and standard deviation across three experimental runs. divAPO consistently delivers the highest win-rate across all datasets, both for the DPO and SLiC-HF loss functions. This demonstrates the effectiveness of divAPO in leveraging a more informed sample selection process, effectively optimizing preference learning and leading to superior fine-tuning performance compared to existing active learning strategies. These results confirm that divAPO not only generalizes well across different datasets but also significantly improves the model's performance by maximizing learning capacity through its novel 1-step selection strategy. This highlights the robustness and effectiveness of our method in preference optimization scenarios, making it a reliable choice for active learning tasks.

## 5.3 ABLATION STUDIES

**Effect of diversity($\gamma$).** The preference model certainty and the diversity of the preference dataset are both essential factors in preference learning. To understand the factors that enhance the performance of divAPO, we assessed it with a variety of diversity parameters, $\gamma \in \{0.0, 0.5, 1.0, 2.0, 5.0, 10.0\}$, to examine the influence of response quality and diversity on the performance of the fine-tuned models. The results in Figure 4 (a) show that divAPO achieves the highest performance with $\gamma = 2.0$, outperforming configurations with either higher or lower $\gamma$ values. We experimented on the IMDB dataset, and subsequently applied this optimal using $\gamma = 2.0$ to experiments of other datasets. The

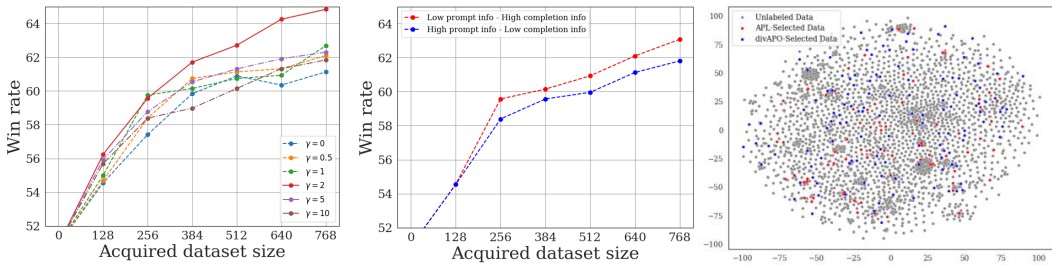

(a) Diversity parameter $\gamma$.     (b) Learning capacity of candidates.     (c) Embedding of APL and divAPO.

Figure 4: Ablation studies. (a) Performance comparison of divAPO with varying diversity parameters showing the best performance at $\gamma = 2.0$; (b) highlights the importance of completion informativeness over prompt informativeness for improving model performance; (c) The distribution of data selected by APL and divAPO methods, showing clustering patterns that suggest potential selection biases in certain data regions.

significance of these factors is underscored by this discovery, which is consistent with prior research that has emphasized the necessity of harmonizing diversity and reward gap in the development of an effective preference dataset.

**Learning capacity of a gray area.** As referenced in Figure 1 (b), we conducted a comparison between two types of data: those with low prompt informativeness but high completion informativeness, and those with high prompt informativeness but low completion informativeness. This experiment was carried out using the IMDB dataset under the DPO loss function. The results in Figure 4 (b) showed a clear trend, data with higher completion informativeness significantly outperformed the data with higher prompt informativeness. This demonstrates that focusing on the completion's informativeness provides more valuable learning signals than relying on prompt informativeness alone. The findings suggest that the strategy of selecting prompts first and generating completions as a secondary step may be sub-optimal, especially in scenarios where the model needs to learn from the richness of the completion data. This supports the hypothesis that informative completions play a crucial role in optimizing preference-based models like DPO, offering a more refined learning process that ultimately leads to superior results.

**Embedding space of selected point.** In the embedding space visualization in Figure 4 (c) using the IMDB dataset under the DPO loss function, we observed an important pattern when applying the 2-step selection method. The selected samples tended to cluster in one particular region of the space, indicating a concentration of data types within a specific category. This region primarily consisted of the input that was either the same or contained difficult content, which was over-represented in the selected dataset. As a result, the GPT model's performance declined, as it did not have sufficient exposure to a broader range of completions. This uneven distribution of selected data highlights a key limitation of the 2-step approach, which can lead to biased or skewed learning outcomes when the diversity of the data space is not adequately covered. This imbalance underscores the importance of ensuring a more even data distribution during the active learning process to prevent such gaps in model performance.

# 6 CONCLUSION

In this work, we introduced divAPO, a novel active learning strategy that addresses the limitations of existing approaches in preference optimization. By simultaneously considering the distance estimated probability and preference model certainty, our method overcomes the shortcomings of traditional two-step selection processes, which often lead to suboptimal data pairings. Our one-step selection strategy ensures that the model learns from a more diverse and informative subset of data, thereby enhancing its overall learning capacity. Extensive experiments on multiple datasets, including IMDB, TL;DR, and HH-rlhf, demonstrated the effectiveness of our approach, with performance improvements of up to 9.21% compared to baseline methods. These findings underscore the importance of balancing diversity and preference model certainty in active learning for preference-based optimization.

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

# Active Preference Optimization via Maximizing Learning Capacity
## (Supplementary Material)

## A  PROOF

We conclude Theorem 4.2 by demonstrating the monotonicity and submodularity of Eq.(4) in A.1 and A.2. This is based on the widely accepted observation that the *monotonicity* and *submodularity* of a combinatorial objective ensure that the greedy selection process results in an objective value that is within $(1 - 1/e)$ of the optimum (Feige, 1998).

### A.1  PROOF OF MONOTONICITY

Monotonicity means that adding an element to the subset $S$ does not decrease the value of the objective function. Formally, it should hold that:

$$f(S \cup \{x_j\}) \geq f(S)$$

The objective function is given as:

$$f(S) = \sum_{x_i \in S} P_d(x^i, L) \cdot R(x_i, y_i^{[1]}, y_i^{[2]})$$

where $P_d(x^i, L)$ is the Distance Estimated Probability, defined as:

$$P_d(x^i, L) = \frac{d_p(e(x_i, y_i^{[1]}, y_i^{[2]}), L)}{\sum_{x_i \in \mathcal{X}} d_p(e(x_i, y_i^{[1]}, y_i^{[2]}), L)}$$

and $R(x_i, y_i^{[1]}, y_i^{[2]})$ is the Preference Model Certainty, defined as:

$$R(x_i, y_i^{[1]}, y_i^{[2]}) = \left| \hat{r}(x_i, y_i^{[1]}) - \hat{r}(x_i, y_i^{[2]}) \right|$$

We calculate the value of the objective function after adding a new element $x_j$ to $S$:

$$f(S \cup \{x_j\}) = \sum_{x_i \in S} P_d(x^i, L) \cdot R(x_i, y_i^{[1]}, y_i^{[2]}) + P_d(x^j, L) \cdot R(x_j, y_j^{[1]}, y_j^{[2]})$$

Since both $P_d(x^j, L) \geq 0$ and $R(x_j, y_j^{[1]}, y_j^{[2]}) \geq 0$ by definition, we have:

$$f(S \cup \{x_j\}) \geq f(S)$$

### A.2  PROOF OF SUBMODULARITY

We are given the following objective function:

$$S^* = \arg \max_{S \subseteq \mathcal{X}:|S|=b} \sum_{x_i \in S} P_d(x^i, L) \cdot R(x_i, y_i^{[1]}, y_i^{[2]})$$

where $P_d(x^i, L)$ is the Distance Estimated Probability, and $R(x_i, y_i^{[1]}, y_i^{[2]})$ is the Preference Model Certainty.

To prove submodularity, we need to show that for any sets $S \subseteq T$, the marginal gain of adding an element $x_j$ to a smaller set $S$ is greater than or equal to the marginal gain of adding $x_j$ to the larger set $T$. Formally, we want to prove the following inequality:

$$f(S \cup \{x_j\}) - f(S) \geq f(T \cup \{x_j\}) - f(T), \quad \text{where} \quad S \subseteq T$$

For the set $S$, the marginal gain of adding $x_j$ is:

$$f(S \cup \{x_j\}) - f(S) = P_d(x^j, L) \cdot R(x_j, y_j^{[1]}, y_j^{[2]})$$

Similarly, for the set $T$, the marginal gain of adding $x_j$ is:

$$f(T \cup \{x_j\}) - f(T) = P_d(x^j, L) \cdot R(x_j, y_j^{[1]}, y_j^{[2]})$$

Both $P_d(x^i, L)$ and $R(x_i, y_i^{[1]}, y_i^{[2]})$ exhibit diminishing marginal returns, which leads to submodularity. Let's analyze both terms:

- Distance Estimated Probability $P_d(x^i, L)$: This term is based on the distance function $d_p$, which measures the distance between point $x_i$'s embedding and the label set $L$. As more elements are added to the set, the relative contribution of each new element to the probability distribution decreases, especially as new elements become more similar to existing ones. Thus, $P_d(x^i, L)$ tends to diminish as the size of the set grows, even though it may not be strictly concave.

- Preference Model Certainty $R(x_i, y_i^{[1]}, y_i^{[2]})$: This term measures the gap between the model's predicted preferences for two different labels. As the set grows and more data is available, the model becomes more confident, and the relative uncertainty or disagreement between the predictions decreases. This leads to a diminishing contribution of additional elements as the set size increases.

Since both $P_d(x^i, L)$ and $R(x_i, y_i^{[1]}, y_i^{[2]})$ are affected by diminishing returns, the marginal contribution of a new element $x_j$ to a smaller set $S$ will be greater than or equal to its marginal contribution to a larger set $T$.

Thus, we can conclude that:

$$f(S \cup \{x_j\}) - f(S) \geq f(T \cup \{x_j\}) - f(T)$$

Since the diminishing marginal returns property holds for both $P_d(x^i, L)$ and $R(x_i, y_i^{[1]}, y_i^{[2]})$, the objective function satisfies the submodularity condition. Therefore, the function exhibits diminishing marginal returns as the size of the subset increases, proving submodularity.

## B  IMPLEMENTATION DETAILS

| Data | IMDB | TL;DR | HH-rlhf |
|---|---|---|---|
| Data size | 25k | 117k | 170k |
| Model used | Pre-trained GPT-2 | Pre-trained Pythia 1b | Pre-trained Pythia 2.8b |
| Optimizer | ADAM lr: 1e-06 | ADAM lr: 1e-06 | ADAM lr: 1e-06 |
| Finetuning Epochs | 30 | 50 | 50 |
| Mini-batch size | 32 | 32 | 32 |
| Prompt query | 128 | 128 | 128 |
| $\beta$ for KL term | 0.2 | 0.2 | 0.2 |

Table 2: Experimental settings for active preference optimization with IMDN, TL;DR, and HH-rlhf.

**Temperature adjustment.** We employ temperature-scaled sampling, which involves scaling the logits before applying the softmax function to modify the probability distribution over the next token. The distribution is sharpened by a low temperature $T < 1$, which results in the model being more conservative and confident in its predictions, which often results in less diverse outputs. The distribution is flattened by a high-temperature $T > 1$, which increases the diversity of the output by

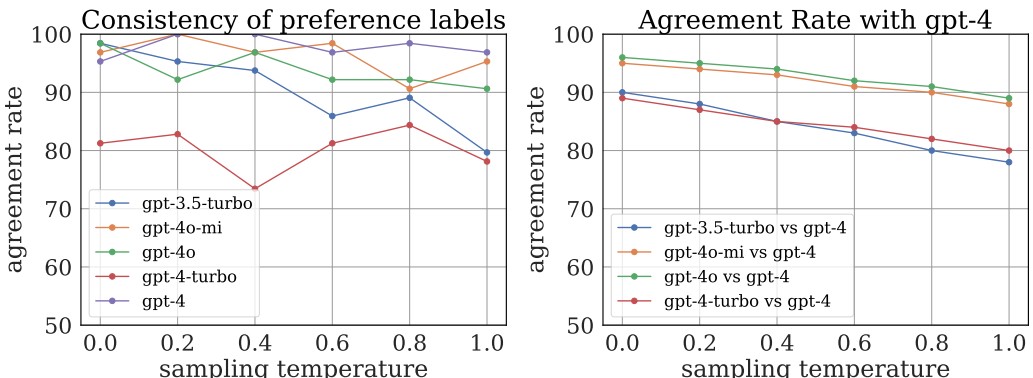

Figure 5: (a) Self-consistency of GPT models. (b) Agreement Rate with GPT-4. (a) shows average self-consistency of preference labels provided by GPT-3.5-turbo, GPT-4o-mini, GPT-4o, GPT-4-turbo and GPT-4 across 64 prompt completion pairs. Each model provided two preference labels for each prompt completion pair. (b) shows agreement rate with GPT-4 across different sampling temperatures. GPT-4o and GPT-4o-mini show slightly higher agreement rates with GPT-4 compared to GPT-3.5-turbo and GPT-4-turbo, especially at lower temperatures.

increasing the likelihood of selecting less probable tokens. The sampling is effectively transformed into greedy decoding when the temperature is zero ($T = 0$). In our experiments, we employ $T = 0.7$ during training, $T = 0.1$ during testing, and $T = 0.05$ for the GPT-4o-mini oracle to encourage reduced variance.

**Data pre-processing** Every input prompt for IMDB is chosen at random at the start of a review. Here, the only processing we perform is to truncate the input prompt at random to a set of tokens, 8–16 tokens, chosen at random. We selected Reddit posts with between 200 and 1000 characters for TL;DR and HH-rlhf The primary cause of this was the GPUs' memory constraints during the model-training process. In the end, we eliminated trailing space tokens.

## C  GPT AS ORACLE EVALUATOR

**Using GPT as Oracle.**

In traditional active learning settings, an oracle is often a human labeler who annotates unlabeled data. For many tasks, this involves providing a clear, singular label, which the oracle then uses to guide the model. However, in tasks requiring preference judgments, where labels are given based on relative evaluation, an immediate and adaptable oracle is needed to handle the latest model-generated responses. Given the impracticality of conducting objective experiments with multiple human judges for every evaluation, we leverage the language model APIs provided by OpenAI for fine-tuning and evaluation processes. Research from the DPO paper indicates that OpenAI's GPT-4, when appropriately prompted, aligns closely with human judgments, demonstrating strong agreement.

**Cost-efficient GPT model.**  One notable drawback of using GPT-4 as the oracle model is its high cost and latency. In response, several alternative models such as GPT-3.5-turbo, GPT-4-turbo, GPT-4o, and GPT-4o-mini have been proposed since the publication of the DPO paper. To determine the most suitable model, we conducted a consistency test using 64 prompts and their associated completions to generate preference labels twice. As depicted in Figure 5, GPT-4o, and GPT-4o-mini exhibit over 90% consistency across 64 data points, with GPT-4 and GPT-4o-mini also demonstrating high response similarity exceeding 90%. Consequently, for a more cost-effective active learning framework, we opted to use GPT-4o-mini as the oracle model.

**Prompt Design.** Utilizing GPT models as oracles necessitates carefully designed prompts tailored to each specific task. Different prompts are employed depending on the nature of the task, as detailed in the appendix. Prompts are evaluated based on their effectiveness in reflecting human preferences, grammatical correctness, functional performance, and consistency of responses, with

selection criteria similar to those used in the APL paper. This approach ensures that the prompts are not only task-appropriate but also optimize the performance of the oracle in generating reliable preference labels.

# D    PROMPT FOR ORACLE GPT

## D.1    IMDB SENTIMENT GPT-4O-MINI WIN RATE PROMPT

```
You are a helpful assistant who evaluates the quality \
and positive sentiment of movie reviews. \
Which of the following movie reviews is better? \
The best one will be the one with the most positive sentiment, \
which also is grammatically correct, consistent, and \
avoids repetition.

Prompt: <prompt>

Review A: <Review A>

Review B: <Review B>

FIRST provide a one-sentence comparison of the two reviews, \
explaining which you prefer and why. SECOND, on a new line, \
state only "A" or "B" to indicate your choice. You must choose \
A or B for the preferred answer even if neither review is \
very good. Your response should use the format:
Comparison: <one-sentence comparison and explanation>
Preferred: <"A" or "B">
```

## D.2    SUMMARIZATION GPT-4O-MINI WIN RATE PROMPT

```
You are a helpful assistant that evaluates the quality of \
summaries. Which of the following summaries does a better \
job summarizing the most important points in the given \
forum post, without including unimportant or irrelevant \
details that are grammatically correct, consistent, and \
avoid repetition?

Post: <post>

Summary A: <Summary A>

Summary B: <Summary B>

FIRST provide a one-sentence comparison of the two summaries, \
explaining which you prefer and why. SECOND, on a new line, \
state only "A" or "B" to indicate your choice. You must choose \
A or B for the preferred answer even if neither summary is very \
good. Your response should use the format:
Comparison: <one-sentence comparison and explanation>
Preferred: <"A" or "B">
```

## D.3    CHATBOT GPT-4O-MINI WIN RATE PROMPT

```
You are a helpful assistant that evaluates the quality of \
chatbot. For the following query to a chatbot, which response \
is more helpful?
```

```
Query: <the user query>

Response A: <either the test method or baseline>
Response B: <the other response>

FIRST provide a one-sentence comparison of the two responses \
and explain which you feel is more helpful. SECOND, on a new \
line, state only "A" or "B" to indicate which response is \
more helpful. Your response should use the format:
Comparison: <one-sentence comparison and explanation>
More helpful: <"A" or "B">
```

## E  LIMITATION AND POTENTIAL NEGATIVE SOCIETAL IMPACT

**Limitation.** While the proposed method offers a novel approach to Active Learning with Preference Optimization, there are several limitations to consider. First, the method's effectiveness is highly dependent on the quality and diversity of the preference feedback. In cases where the feedback is biased or incomplete, the model's learning capacity may be restricted, potentially resulting in suboptimal performance. Second, the 1-step selection strategy, while efficient, may struggle to scale effectively in extremely large datasets due to computational constraints, particularly when calculating distance-estimated probabilities (DEP) and reward gaps across massive data pairs. Additionally, while our approach maximizes learning capacity for a specific task, it may not generalize well to tasks with highly distinct structures or feedback modalities, limiting its broader applicability.

**Potential Negative Societal Impact.** The widespread application of Active Learning with Preference Optimization could have unintended societal consequences. By prioritizing human preferences, models trained using this method might inadvertently reinforce biases present in the feedback data, especially in cases where feedback reflects socially biased or exclusionary viewpoints. If such models are deployed in sensitive applications, such as recommendation systems or content moderation, they could perpetuate or exacerbate inequalities, including discrimination based on race, gender, or socioeconomic status. Furthermore, reliance on human feedback at scale may introduce ethical concerns regarding labor exploitation, particularly in crowdsourced settings where annotators are compensated at low wages. These societal impacts highlight the need for careful consideration of feedback sources and post-hoc auditing of models trained using AL-PO methods to mitigate harmful biases and ensure ethical deployment.

