# OpenReview forum: "Active Preference Optimization via Maximizing Learning Capacity"
_ICLR.cc/2025/Conference — ICLR 2025 Conference Withdrawn Submission_

### Official Review · Reviewer_PScs · 2024-10-30

**Soundness:** 3
**Presentation:** 1
**Contribution:** 2
**Rating:** 3
**Confidence:** 4

**Summary:**

The paper presents a novel active learning algorithm divAPO in preference learning.
A distance-estimated probability is proposed to compute the expected certainty.
For reasonable computation complexity, greedy algorithm is applied to iteratively select data with theoretical guarantess.
Empirical evaluation is conducted to align GPT-2, Pythia-1B, and Pythia-2.8B on IMDB, TL;DR, and Anthropic Helpful and Harmless datasets using DPO and SLiC-HF respectively, where divAPO achieves consistently higher win rate where GPT-4o-mini serves as the judge.

**Strengths:**

* The paper addresses an important problem, i.e., data selection in preference learning.
* In the empirical evaluation, the proposed method consistently outperforms the baselines.

**Weaknesses:**

The paper is not clear to me.
For example, in Algorithm 1
* It seems only $y_i^{[1]},y_i^{[2]}$ are involved in the following procedure.
Why sampling $y_i^{[3]},\dots,y_i^{[n]}$ on line 3?
* Set $L$ is not initialized or populated anywhere, but referenced on line 6.
What elements does $L$ contain?
* There is summation over $k$ on line 6, but $k$ does not occur in the summed term.
* Do $(x_i,y_i^{[1]},y_i^{[2]})$ and $(x^i, y_1^i, y_2^i)$ refer to the same tuple?
If so, the notation should be unified.
* On line 9, argmax is applied to prompt $x$.
How does this lead to tuple $(x_*,y_*^+,y_*^-)$?
* The elements in $U$ are prompts $x$ and $\mathcal S$ are tuples $(x_*,y_*^+,y_*^-)$.
How does the operation $U\backslash \mathcal S$ work?
* I understand line 9 calls the oracle $\mathcal O$ to obtain the preference, which should be explicitly stated.
* Set $S$ is populated on line 11 but the terminal condition is on line 10.
To my understanding the inner loop is an infinite loop?

I encourage the authors to revise the paper for the ease of understanding.

**Questions:**

* The embedding function $e$ encodes a tuple $(x_i,y_i^{[1]},y_i^{[2]})$.
What is the format of the input?

---

### Official Review · Reviewer_Fee1 · 2024-10-30

**Soundness:** 2
**Presentation:** 3
**Contribution:** 2
**Rating:** 5
**Confidence:** 4

**Summary:**

This work seeks to address the problem of active learning for RLHF fine-tuning. Specifically, the authors posit that prior approaches fail to consider the informativeness of prompt and response pairs together, instead opting for two-step selection methods. The work proposes divAPO, a single step algorithm which jointly selects prompt, response pairs for preference learning. divAPO is tested on the IMDB, Tl;DR, and HH datasets using the DPO and SLiC-HF.

**Strengths:**

This work addresses an existing problem in the setting of active learning for preference optimization, they build on previous works to suggest an approach that accounts for the informativeness of the model prompt and responses in a one-step selection process.

The writing and motivation of the paper are clear. Most of the technical details are explained in a manner that is easy to understand. The authors include code for reproducing the experiment results. The experimental results presented in the paper are strong and statistically significant when compared to relevant baseline motivating the application of divAPO in practical settings.

**Weaknesses:**

The weaknesses of the paper are as follows:

1.	Missing references in the related works. [1] introduce regret bounds for online iterative learning with batch exploration in the RLHF problem.

2.	The proof of sub modularity (paragraphs beginning at lines 716 and 721) in the appendix should not be presented as a proof. The arguments the authors present are reasonable but are not technical proofs of submodularity. The submodularity of the Preference Model Certainty term specifically is likely only approximate as the implicit reward relies upon the LLM policy. In the main text the submodularity property of the selection objective should therefore be presented as an assumption before Theorem 4.2. The current presentation may be interpreted as misleading.

3.	Some technical details are missing from the paper. These include:

-	A brief introduction to the k-means ++ cluster approach, specifically introducing gamma as this is key to understanding Figure 4a), and the effect of k more broadly.

-	How the embedding space is implemented in the experiments. Are the prompts and response embeddings added or concatenated together, or is some other approach used?

-	How are the 2D embeddings in Figure 4c) created?

-	In algorithm 1 line 3 multiple responses are generated per prompt $y^{[n]}$. Should this just be two responses or are there n responses?

[1] Wei Xiong, Hanze Dong, Chenlu Ye, Ziqi Wang, Han Zhong, Heng Ji, Nan Jiang, and Tong Zhang. Iterative preference learning from human feedback: Bridging theory and practice for rlhf under kl-constraint. In International Conference on Machine Learning, 2024.

**Questions:**

I have a few broader questions about the approach and the results:

1.	What does divAPO stand for?

2.	The experiment results presented still show the model results improving, if a larger dataset is collected do the baseline approaches eventually achieve the same performance as divAPO or does their win-rate plateau below divDPO?

---

### Official Review · Reviewer_eJc5 · 2024-11-04

**Soundness:** 3
**Presentation:** 2
**Contribution:** 2
**Rating:** 5
**Confidence:** 4

**Summary:**

The paper introduces divAPO, a method for active preference optimization to reduce the annotation burden of RLHF. The authors argue that divAPO eliminates suboptimal pairings and enhances learning capacity. Experiments on language tasks show that divAPO outperforms existing approaches.

**Strengths:**

1. The question of the paper is interesting and useful to the field of model alignment.
2. The experiment results are promising and show the effectiveness of the proposed method.

**Weaknesses:**

1. My main concern is the relationship between the proposed method and the previous work. In section 2, the authors mentioned the previous AL algorithms "hard to directly apply to the PO framework" as "only a single label is required for annotating each data example". However, in section 3.3, the authors are able to convert the pair of labels to a single certainty label. In this case, would the previous work applicable to the preference optimization framework? If yes, a thorough discussion of the applicability of the traditional AL algorithms would be helpful.
2. The presentation of the work has room for improvement. This includes the unclear definition of terms, the lack of motivation for the proposed method, and overuse of the language to describe the mathematical operations.
- Unclear definition of terms: Please see the questions part for detailed comments.
- Insufficient motivation: The authors did not provide a clear motivation for eq 3, where the selection is by the multiplication of two terms that the authors believe important. How about other operations like addition?
- Overuse of the language: line 266, line 291. The authors can easily use the mathematical formulation or refer to some equations to help the reader understand the operations.
3. As mentioned in line 393, "OpenAI’s GPT-4, when appropriately prompted, aligns closely with human judgments". In this case, it is not clear why the  a significant number of responses from human participants (line 35, the motivation of the work) is necessary and a burden, as strong model like GPT-4 or open-sourced counterpart LLama can provide preference labels.

**Questions:**

1. In line 70, the authors mentioned that "input prompts that contain less frequently encountered in everyday use" are suboptimal. It is not clear to me why we need to pay less attention to such less common content. In opposite, sometimes the models are blamed for their lack of understanding of some uncommon and complex content like the philosophical content. Could you please provide more explanation on this point?

2. Some terms are used without clear definitions which make it hard to follow the paper.
- line 078, what is the "2-step selection"?
- line 086, how do you define "sub-optimal candidate"?
- line 213, what is $P(\cdot)$ and why this function can be applied to both the dataset $U$ (line 161, 213) and the data point $x_i$ (eq 3)?
- line 249, $d_p(\cdot)$ represents the distance. But what is $d_p(\cdot,\cdot)$ as in eq 5
- line 426, what is $\gamma$

---

### Note · Authors · 2024-12-19

I have read and agree with the venue's withdrawal policy on behalf of myself and my co-authors.